# Neonatal Meningitis Due to Herpes Simplex Virus Type 1 and Enterovirus Coinfection: Case Report and Review of the Literature

**DOI:** 10.3390/v17060790

**Published:** 2025-05-30

**Authors:** Carolina Coramusi, Alessandra Rizzoli, Caterina Alegiani, Susanna Bonafoni, Cinzia Auriti, Pasquale Parisi, Maria Eleonora Scapillati

**Affiliations:** 1Faculty of Medicine and Psychology, Sapienza University, 00185 Rome, Italy; 2San Pietro Fatebenefratelli Hospital, Neonatal Intensive Care Unit, Sapienza University, 86100 Rome, Italy; 3Departmental Faculty of Medicine, Saint Camillus International University, 00131 Rome, Italy; 4NESMOS Department, Faculty of Medicine & Psychology, Sapienza University, c/o Sant’ Andrea Hospital, 00185 Rome, Italy

**Keywords:** neonatal meningitis, HSV, enterovirus, coinfection, newborn, case report

## Abstract

In the first 90 days, most meningitis cases are viral. Newborns often show nonspecific symptoms, making early diagnosis difficult but crucial for effective treatment and good outcomes. Cerebro-spinal fluid (CSF) analysis is the gold standard for diagnosis, enabling targeted therapy. We report on a newborn with rare viral meningitis due to herpes simplex virus type 1 and enterovirus coinfection. This uncommon situation complicates diagnostic and therapeutic management. We share our experience and review the limited literature on such neonatal viral coinfections.

## 1. Introduction

Newborns are highly susceptible to infections due to their immature immune systems, lacking immunological memory and adaptive immunity [1].

Viruses cause about two-thirds of central nervous system (CNS) infections in the first 90 days of life. Enteroviruses (EVs) mainly result in self-limited aseptic meningitis with no long-term effects, having an incubation period of 3–6 days [2,3]. Delayed diagnosis and treatment of herpes simplex virus (HSV) CNS infections can lead to severe morbidity and mortality [2].

### 1.1. Enteroviruses

EVs are common infectious agents in humans, causing diseases from the common cold to polio and aseptic meningitis. They infect the human alimentary tract and belong to the Picornaviridae family, including enterovirus (A–J), coxsackievirus, poliovirus, echovirus, and rhinovirus (A–C). Non-polio EVs cause febrile illnesses in newborns, especially in summer, and can lead to multisystem diseases, sometimes fatal. Symptoms range from asymptomatic viral shedding to severe conditions like aseptic meningitis and myocarditis. Most infections in newborns are benign, but timely diagnosis is essential for effective management.

### 1.2. Herpes Simplex Viruses

There are two types of HSV: herpes simplex type 1 (HSV-1) and herpes simplex type 2 (HSV-2). HSV-1 typically causes oral herpes, while HSV-2 is known for genital herpes, spread through sexual contact. Lately, HSV-1 has been closely associated to genital infections [4].

Though both HSV-1 and HSV-2 can affect pregnant women, neonatal infection remains rare, with an incidence of 1 in 10,000 live births in Europe, mostly due to HSV-1 [5]. Seroprevalence varies by age, sex, race, and location. Most infections are asymptomatic during pregnancy [5].

Genital herpes can be transmitted to the newborn if active lesions are present at birth, especially during primary maternal infection close to delivery, with a transmission risk of 50–60%, compared to less than 3% for recurrent infections. This higher risk is due to the absence of transplacental antibodies and increased viral exposure in the birth canal [5,6].

Fortunately, most genital herpes infections during pregnancy are recurrent, with a much lower risk of neonatal transmission [5,6,7].

Neonatal infections can manifest as skin, eye, and mouth (SEM) infections, generalized sepsis, or meningoencephalitis, often with severe outcomes.

This report presents a rare case of neonatal meningitis caused by coinfection of HSV-1 and EV. A literature review was conducted to understand the epidemiology, clinical presentation, diagnosis, and treatment for these infections; therefore, we share the experience of our department with the intention of offering practical guidance and support to colleagues who may encounter these rare cases, helping to improve clinical management and patient outcomes.

## 2. Methods

A literature search was conducted to include the most relevant studies on HSV-1, EV infections in newborns, and their impact on newborns when contracted during pregnancy. We explored the following databases: PubMed/MEDLINE, Scopus, Web of Science, and Google Scholar.

The literature on this topic is limited, and we included articles published from January 2005 to June 2024. The search terms included the following keywords: “Enterovirus, EV in newborns, HSV-1 AND newborns, Herpes simplex 1-2, HSV maternal infection, fetal outcomes, neonatal infection, vertical transmission, congenital infection”. We included studies that focused on either HSV or EV meningitis separately, as cases describing coinfection are rarely reported in the literature. The time frame from January 2005 to June 2024 was chosen to ensure the inclusion of up-to-date evidence, given the significant advances in both diagnostic techniques and therapeutic strategies over the past two decades.

## 3. Case Presentation

A male, newborn five-day-old was brought by his parents to the emergency room (ER) because of the onset of fever in the previous 24 h. He was born at term via spontaneous vaginal delivery, following an uneventful pregnancy, with no significant obstetric history. The newborn had a normal rooming-in period and an appropriate birth weight, and he was discharged at 48 h of life and exclusively breastfed. The mother reported experiencing fever, common cold symptoms, and occasional diarrhea during the two days preceding delivery.

Upon admission to the ER, the neonate’s axillary temperature was 38.7 °C, and he had a heart rate of 170 bpm, blood pressure of 65/45 mmHg, and SaO_2_ of 99% in 0.21% FiO_2_. The infant was in good general condition and eupneic, with mild irritability and consolable crying; his cardiothoracic examination was normal, and his neonatal reflexes were present. Despite being febrile and without meningeal signs, the infant had shown reduced feeding in the preceding hours due to breast milk refusal.

Laboratory tests demonstrated neutrophilic leukocytosis (13.6 × 10^9^/L/mmc) and elevated inflammatory markers: *C-reactive protein–CRP*: 26 mg/L (normal value < 5 mL/L); *Procalcitonin—PCT:* 4.75 ng/mL (normal value < 0.5 ng/mL). Liver, kidney, and coagulation profiles, blood glucose levels, and arterial blood gas analysis were normal. The newborn was admitted to our Neonatal Intensive Care Unit, and urine and blood cultures were obtained during a febrile episode. Broad-spectrum antibiotic therapy was started with an ampicillin 100 mg/kg/dose every 12 h and gentamicin 4 mg/kg/die.

Six hours after admission, the infant presented hypotonia and hyporeactivity. Due to persistent fever, we performed a lumbar puncture (LP). The cerebrospinal fluid (CSF) examination indicated normal chemical–physical findings, a negative microbiological culture, and positive polymerase chain-reaction (PCR) results for EV RNA and HSV-1 DNA. Molecular testing showed positive results for EV RNA in plasma and stool but negative results in rectal, oral, and conjunctival swabs. A diagnosis of congenital meningitis due to EV and HSV-1 was established, based on the short period between birth and the onset of the clinical symptoms in the newborn.

Intravenous human immunoglobulins (IVIG) enriched with IgM (*Pentaglobin ev fl 50 mg/mL 10 mL BioTest Italia*) were administered for three days, in conjunction with intravenous acyclovir at a dosage of 20 mg/kg every six hours. Twenty-four hours later, the infant began to exhibit gradual clinical improvement. The recovery of age-appropriate muscle tone was noted, and neurological examinations indicated development consistent with normal parameters. The fever subsided, and laboratory tests demonstrated a progressive reduction in inflammatory markers. Blood cultures at the 72 h mark returned negative results.

During the hospitalization, additional tests were performed to complete the diagnostic evaluation:-*Electroencephalogram (EEG)*: no electroclinical seizures, age-appropriate organized tracing.-*Cerebral MRI with contrast*: no pathological findings in brain tissues.-*Neurological examination*: no significant neurological abnormalities.-Auditory brainstem response (ABR), ECG, and echocardiography: all returned normal results.

A further investigation of the maternal history revealed positive HSV-1/2 IgM serology, while the IgG level was negative. This was accompanied by a clinical history and physical examination that were otherwise unremarkable.

After 21 days of therapy, a repeated PCR on CSF was negative. The antiviral therapy was continued intravenously for a total of 21 days, followed by oral administration for six months, with a dosage of 300 mg/m^2^ per dose, administered three times daily. The patient was discharged from our unit at 30 days of life in good general condition, afebrile, eupneic, and exclusively breastfed. Neurological follow-up showed neurodevelopmental outcomes consistent with the infant’s age.

Currently, the patient is 22 months old. Both neurological examinations and instrumental evaluations are within normal limits. At 2 months of age, the patient underwent second-level hearing screening, including transient evoked otoacoustic emissions (TEOAEs), which showed a bilateral PASS result, and automated ABR, which also showed a bilateral PASS at 35 dBnHL. These findings are indicative of normal cochlear function and auditory pathway integrity up to the brainstem on both sides.

The patient underwent regular follow-up in pediatric infectious diseases until the age of one year, including serial cranial ultrasounds, which showed no abnormalities, as well as neurological, neuropsychological, and psychological assessments, all of which documented normal psychomotor development. The Infant and Toddler Developmental Bayley Scales, Third Edition (Bayley-III, were administered, showing developmental scores within normal limits.

Serial blood tests were also performed, demonstrating past immunity to HSV (positive IgG and negative IgM), negative PCR for enterovirus, and no abnormalities in hepatic or renal function. The patient received prophylactic treatment with oral acyclovir at a dose of 0.9 mL every 8 h, continued until 6 months of age. The next scheduled follow-up is planned at two years of age.

## 4. Discussion

### 4.1. Neonatal HSV-1 Infection

Neonatal HSV infection can occur in utero (5% of cases), peripartum (85% of cases), or postpartum from family members and caregivers (10% of cases) [5,6,7].

Congenital transmission can happen with both primary and recurrent maternal infections but is rare. Symptoms at birth often include skin lesions, scarring, neurological issues like microcephaly and intracranial calcifications, and ocular signs such as chorioretinitis and optic atrophy [7,8].

Globally, infections with HSV- 1/2 are more frequently associated with long-term neurodevelopmental impairments and higher mortality rates compared to EV infections. Data on the clinical features and long-term neurological outcomes of neonatal CNS coinfection with these two viruses are limited. In this case, the patient presented with early-onset meningitis, likely of congenital origin, resulting from dual viral infection. Only maternal serology for HSV-1 was available, as current obstetric protocols do not include routine vaginal swab screening for HSV-DNA prior to delivery.

A vaginal swab test for HSV conducted on the mother during the patient’s hospitalization yielded a negative result. It was noted that the mother had diarrhea and fever at the time of delivery. The therapy was initiated promptly, and the baby tolerated it well, without reporting any side effects or neurodevelopmental sequelae to date.

A favorable outcome is generally observed in EV meningitis in term newborns, with adverse neurological sequelae rarely reported [9]. In contrast, HSV-1 infections of the CNS are associated with a less favorable prognosis. They represent 5% to 15% of all infectious encephalitis cases in children and adults and tend to have particularly severe consequences in the neonatal period [10].

Over the past decade, the use of PCR on CSF samples has significantly improved the diagnosis of CNS infections, while the introduction of antiviral therapies has helped reduce disease severity. Before the adoption of prolonged high-dose antiviral treatment for six months, neonatal herpes infections had a mortality rate of around 30%; today, this has decreased to approximately 10% [11].

Therefore, when faced with a febrile newborn, in addition to conducting culture tests on blood, urine, and other biological fluids, when necessary, it is recommended to always perform an LP to search for bacteria and viruses using molecular biology methods. The overlap in early symptoms of bacterial and viral meningoencephalitis, along with a possible lack of CSF pleocytosis in HSV, underscores the importance of molecular diagnostic panels to ensure timely and accurate treatment [12,13].

CSF PCR has a sensitivity ranging from 75% to 100% for detecting neonatal HSV infection; however, a negative result does not exclude central nervous system involvement, especially in the presence of abnormal CSF parameters, neuroimaging findings, EEG abnormalities, or clinical seizures [2,3,6,12].

For our patient, a positive CSF PCR for HSV-DNA indicates CNS involvement. However, a positive blood culture for HSV-DNA does not quantify symptom severity (SEM, CNS, or disseminated) [5,6,14,15]. Moreover, if the clinical presentation is suggestive of sepsis and cultures are pending or negative, empiric antiviral therapy should be initiated—even in the absence of maternal history or positive serology [16,17].

For pregnant mothers, HSV-1 and HSV-2 are no longer limited to labial or genital infections. HSV-1, once mainly oral, and HSV-2, traditionally genital, can now occur in either region. HSV-1 increasingly causes genital cases, usually with milder symptoms than HSV-2. Researchers predict that, by 2050, up to 25% of HSV-1 infections in the U.S. will be genital [18,19]. A recent prospective study conducted in Australia found that HSV-1 is the primary cause of the first episodes of genital herpes in many countries. The study examined viral shedding in the saliva and genital secretions of 82 patients (65.9% women) after a laboratory-confirmed first HSV-1 infection to measure its persistence after symptom disappearance. HSV-1 was detected in the genital tract in 64.6% of patients and in the mouth in 29.3% of patients two months post-infection. Genital shedding of HSV-1 significantly decreased 11 months after the onset of infection. Overt genital lesions were uncommon, occurring on only 3.8% of examination days at that time point. These findings suggest that primary HSV-1 genital infections in pregnant women may result in prolonged viral shedding, with potential for active transmission through genital secretions for up to 11 months following initial infection. Consequently, a patient with an HSV-1 infection contracted during the second trimester of pregnancy may be contagious to the newborn at birth, regardless of symptom presence. In cases of early febrile infections in newborns, it is crucial to examine maternal medical and virological history and conduct molecular testing for HSV-1/2 in the baby’s blood and CSF. If bacterial coinfection is excluded, a positive PCR result can prevent unnecessary prolonged antibiotic therapy and hospital stay [16,18,20].

### 4.2. Neonatale EV Infection

EVs often infect newborns, with coxsackie B and echoviruses causing the most severe cases, which are frequently underdiagnosed. An Italian study found that 11.6% of 60 newborns with sepsis-like symptoms had EVs, a rate similar to that in the U.S. [21,22].

Despite this, EV testing is not standard for early-onset infections. Including it could reduce antibiotic use, hospital stays, and healthcare costs and allow early patient isolation to prevent virus spread [14,15]. One of the primary motivations for writing this article was to highlight the importance of considering EV testing in neonatal infections, particularly to minimize the empirical use of antibiotics.

The incubation period for EV infections is 3–6 days [3]. After entering through the digestive or respiratory tract, EVs initially replicate locally in the pharynx or gastrointestinal tract and then spread via minor viremia to other sites like the CNS, heart, liver, respiratory system, or skin, leading to major viremia and potential CNS infection [15].

Most symptomatic infants show nonspecific fever (30%) or aseptic meningitis (40%) with quick recovery. However, about 30% of infected newborns, especially preterm infants, may develop severe complications such as fulminant hepatitis, myocarditis, or multi-organ failure (Table 1) [15,23,24,25].

Recent advancements in viral diagnostics have revealed an uptick in severe neonatal infections caused by Echovirus 11 (E11) [2]. This trend has been reported by French authorities through the European Monitoring System from July 2022 to 2023. It has been observed that viruses isolated in Italy belong to a new lineage, potentially leading to more severe infections [26]. These data indicate that EV infections in newborns can sometimes result in serious conditions.

The exact incidence of this disease is difficult to establish. EVs are highly contagious and environmentally stable, capable of surviving for several days at room temperature and withstanding gastric acidity, making them well suited via the fecal–oral route [3]. Maternal–fetal EV transmission may occur antenatally, perinatally, or postnatally. Typically, virus transmission to the newborn happens through contact with infected maternal secretions during vaginal delivery [27,28]. Transplacental transmission of EVs is also possible and is generally associated with more severe, sometimes fatal, infections. In these cases, symptoms typically appear within the first few days of life. Antenatal transmission, whether transplacental or ascending, can be confirmed by detecting positive viral cultures from throat or rectal swabs within the first three days of life, as well as through the presence of blood’s IgM before five days. Additionally, the virus may be detected in amniotic fluid, umbilical cord blood, and tissues in cases of severe infection resulting in in utero death [29].

Identifying the exact route of neonatal EV infection can be difficult; however, maternal illness and the timing of symptom onset in the newborn can offer important clues. When the mother develops symptoms within two weeks prior to delivery, she is likely the source of the infection. Given that the incubation period for enteroviral disease typically ranges from 3 to 5 days (with a broader range of 2 to 12 days), the appearance of symptoms within the first 48 h of life suggests a transplacental mode of transmission [30].

Some studies suggest that transmission through breastfeeding is also possible. Most neonates with EV have mothers or family members exhibiting viral illness symptoms around the time of delivery, as observed in our case. These newborns are typically full-term and experience uncomplicated postnatal courses before the disease onset.

### 4.3. Our Experience

In our specific case, the newborn had a standard hospital stay with rooming-in arrangements with his mother until discharge, coming back to the hospital on the fifth day due to the onset of fever. Identified risk factors for symptomatic neonatal EV infection include prematurity, low birth weight, being a twin, oropharyngeal procedures, nasogastric intubation or feeding, the need for intensive care, and close contact with the index case [3,15]. Mortality rates are highest among newborns with severe hepatitis, myocarditis, or both conditions combined [3].

Empirical therapy for infants with EV involves IVIG at 750 mg/kg for 3–6 days or hyperimmune immunoglobulin at 250 mg/kg/day for 3 days. Immune globulins support treatment by reducing viremia, viruria, and decreasing morbidity and mortality [12,16,31,32,33].

Leukocyte interferon is also recommended by some authors. These treatments are not used in maternal infection during pregnancy and cannot prevent fatal outcomes [34].

Specific antiviral therapy like pleconaril [35] shows promise in treating meningitis and life-threatening infections by inhibiting viral capsid and preventing RNA release into host cells. In case studies, pleconaril led to full recovery in severe hepatitis cases and substantial improvement in others [36] (Table 2).

## 5. Conclusions

Ultimately, our patient’s case illustrates the significant risks posed by both HSV-1 and EV infections in newborns, necessitating hospitalization and extended inpatient and outpatient care.

EV are widely distributed and can lead to serious consequences in cases of vertical transmission, such as myocarditis and meningoencephalitis.

HSV-1 infections, although infrequent, pose a particular challenge in situations involving asymptomatic maternal infections, where diagnosis depends on advanced methods like RT-PCR.

Associated morbidity includes adverse fetal outcomes such as preterm birth and neurological impairments. In newborns presenting with early fever, it is essential to perform an LP and CSF analysis to detect viral infections. Prompt diagnosis allows for early treatment, which improves clinical outcomes.

The primary treatment for HSV-1 infections is acyclovir, while enteroviral infections are managed through supportive therapy and experimental antiviral treatments like pleconaril.

It is imperative to stress the importance of preventive measures to control HSV-1/2 transmission during pregnancy. Although significant progress has been made in understanding the pathophysiology and transmission of these viruses, there remain substantial gaps in knowledge, particularly concerning the incidence and long-term effects of congenital infections. Improving diagnostic, therapeutic, and preventive strategies is essential for the effective management of this important public health concern.

In the end, we hope this article succeeds in its aim to provide guidance for the management of these rare cases of neonatal meningitis coinfection, offering a useful reference for clinicians when facing such uncommon but clinically significant scenarios.

## Figures and Tables

**Table 1 viruses-17-00790-t001:** Main symptoms of neonatal CNS infection, with the reported frequency of presentation; these symptoms are non-specific and should prompt further investigations to achieve an accurate diagnosis.

Symptom	Frequency
Fever	~90% of cases
Irritability	~70–80%, particularly in the first week of life
Pleocytosis (CSF)	~30–40%, may mimic bacterial meningitis
Hypotonia	~20–30%, varying in severity
Peripheral hypertonia	Rare, observed in isolated cases
Mottled skin (marbled)	~10–15%, often with fever and jaundice
Absence of specific symptoms	Presentation varies; non-specific symptoms in ~20–30%

**Table 2 viruses-17-00790-t002:** Different pharmacological treatments available for viral meningitis according to recent literature.

Therapy	Dosage	Notes
Intravenous immunoglobulin (IVIG)	750 mg/kg for 3–6 days	Supportive therapy; may reduce viremia and viruria, decreasing morbidity and mortality.
Hyperimmune immunoglobulin	250 mg/kg/day for 3 days	Like IVIG; supportive, not curative.
Leukocyte interferon	Not standardized	Recommended by some authors; efficacy not fully established.
Pleconaril	Not standardized; case-specific	Capsid inhibitor; prevents viral RNA release. Promising in severe cases, but limited data available.
Acyclovir	20 mg/kg every 8 h for 14–21 days	Empiric antiviral therapy, particularly if HSV coinfection is suspected.

## Data Availability

No new data were created or analyzed in this study. Data sharing is not applicable to this article.

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
