# Peer review of "Neonatal Meningitis Due to Herpes Simplex Virus Type 1 and Enterovirus Coinfection: Case Report and Review of the Literature"

_viruses, 2025, doi:10.3390/v17060790_

Round 1

Reviewer 1 Report

Comments and Suggestions for Authors

This manuscript presents a rare case of neonatal meningitis caused by co-infection with Herpes Simplex Virus type 1 (HSV-1) and Enterovirus (EV) in a term newborn. The authors describe the clinical presentation, diagnostic approach, treatment, and favorable outcomes of this unusual case. They also provide a literature review on neonatal HSV and enterovirus infections, discussing epidemiology, transmission routes, clinical manifestations, diagnosis, and treatment options.

The case report is well-documented, and the literature review provides a useful context for understanding the epidemiology, clinical presentation, and management of such infections. However, several areas require improvement. Please see my comments:

  1. The literature review methodology is insufficiently detailed. The authors state they searched multiple databases but do not provide specific search strategies, inclusion/exclusion criteria, the number of studies identified and included or quality assessment of selected studies. This limits the reproducibility and rigor of the review. The time frame for the literature review (January 2005-June 2024) is mentioned, but the rationale for this specific period is not explained.
  2. Citations are necessary for both tables to properly attribute the source of the information. Table 1 presents symptoms of "neonatal central nervous system infection" generally, rather than specifically for HSV/EV co-infection, making it less relevant to the unique aspects of this case.
  3. The discussion lacks a structured comparison between this case and similar published cases of HSV/EV co-infection in neonates.
  4. The manuscript would benefit from clearer organization, particularly in the discussion section where information about HSV and EV infections is sometimes intermingled.

Author Response

R: This manuscript presents a rare case of neonatal meningitis caused by co-infection with Herpes Simplex Virus type 1 (HSV-1) and Enterovirus (EV) in a term newborn. The authors describe the clinical presentation, diagnostic approach, treatment, and favorable outcomes of this unusual case. They also provide a literature review on neonatal HSV and enterovirus infections, discussing epidemiology, transmission routes, clinical manifestations, diagnosis, and treatment options. 

The case report is well-documented, and the literature review provides a useful context for understanding the epidemiology, clinical presentation, and management of such infections. However, several areas require improvement. Please see my comments: 

1. The literature review methodology is insufficiently detailed. The authors state they searched multiple databases but do not provide specific search strategies, inclusion/exclusion criteria, the number of studies identified and included or quality assessment of selected studies. This limits the reproducibility and rigor of the review. The time frame for the literature review (January 2005-June 2024) is mentioned, but the rationale for this specific period is not explained. 

2. Citations are necessary for both tables to properly attribute the source of the information. Table 1 presents symptoms of "neonatal central nervous system infection" generally, rather than specifically for HSV/EV co-infection, making it less relevant to the unique aspects of this case. 

3. The discussion lacks a structured comparison between this case and similar published cases of HSV/EV co-infection in neonates. 

4. The manuscript would benefit from clearer organization, particularly in the discussion section where information about HSV and EV infections is sometimes intermingled. 

A: Thank you very much for your valuable feedback and for your careful review of our manuscript. 

As per your suggestions: 

1. We have specified the details of our literature search more clearly in the revised manuscript. 

2. The tables were entirely created by us based on the comprehensive analysis of the literature reviewed; therefore, we are unable to include specific references. However, we have clarified that the symptoms listed in Table 2 are general signs of central nervous system infection, which should be considered warning signs prompting an early diagnostic workup. 

3. We have restructured the Discussion section to improve clarity by dividing it into well-defined paragraphs. 

4. We also performed a thorough and careful English language revision throughout the manuscript. 

We hope that these changes have significantly improved the quality of our work. We look forward to your feedback and remain at your disposal for any further revisions. 

Reviewer 2 Report

Comments and Suggestions for Authors

This manuscript needs to be rigorously revised in terms of English writing. As a systematic report associated with HSV-1 and EV coinfection, there should be a list of patients informations in this manuscript. Of course, the conclusion demonstrates that the enhancing diagnostic, therapeutic, and preventive strategies have  the importance and significance as well as being imperative and so on. What is the scientific value of this academic article? 

There are many minor errors that need to be corrected, and ome of the representative mistakes have been marked out. The paragraph arrangement of this article needs to be reorganized. It is too chaotic and the overall structure of the article is not coherent.

Comments on the Quality of English Language

This manuscript should be considered for revision and editing by native English speakers who are colleagues or editors. Otherwise, there is a risk of deviation in understanding.

Author Response

R: This manuscript needs to be rigorously revised in terms of English writing. As a systematic report associated with HSV-1 and EV coinfection, there should be a list of patients informations in this manuscript. Of course, the conclusion demonstrates that the enhancing diagnostic, therapeutic, and preventive strategies have  the importance and significance as well as being imperative and so on. What is the scientific value of this academic article?

There are many minor errors that need to be corrected, and ome of the representative mistakes have been marked out. The paragraph arrangement of this article needs to be reorganized. It is too chaotic and the overall structure of the article is not coherent. 

A:  Thank you very much for your valuable suggestions, which have been essential in improving our work.

As recommended, we have carried out a thorough English language revision and reorganized the Discussion section into paragraphs to enhance clarity and readability. We have also expanded the explanation of the aim of our review, which is to offer an additional literary resource to support colleagues who may encounter rare clinical cases such as neonatal meningitis caused by HSV and EV. 

We sincerely hope that these revisions have contributed to a significant improvement in the quality of our manuscript. We look forward to receiving your feedback and remain fully available for any further modifications. 

We would also like to express our gratitude for providing the annotated PDF file with your suggestions — a clear sign of your dedication and attention as reviewers, which we truly appreciate. 

Round 2

Reviewer 1 Report

Comments and Suggestions for Authors

The authors have answered all my comments.

Reviewer 2 Report

Comments and Suggestions for Authors

I have no additional comments.